# Body Mass Index during Gluten-Free Diet in Patients with Celiac Disease

**DOI:** 10.3390/nu15163517

**Published:** 2023-08-10

**Authors:** Zsófia Vereczkei, Tímea Dergez, Zsuzsanna Fodor, Zsolt Szakács, Judit Bajor

**Affiliations:** 1Institute for Translational Medicine, Medical School, University of Pécs, 7624 Pécs, Hungary; vereczkei47@gmail.com; 2Department of Sport Nutrition and Hydration, Institute of Nutritional Science and Dietetics, Faculty of Health Sciences, University of Pécs, 7621 Pécs, Hungary; 3Institute of Bioanalysis, Medical School, University of Pécs, 7624 Pécs, Hungary; timea.dergez@aok.pte.hu; 4Medical School, University of Pécs, 7624 Pécs, Hungary; fzsuzsanna99@gmail.com; 5First Department of Medicine, Medical School, University of Pécs, 7624 Pécs, Hungary; szaki92@gmail.com

**Keywords:** celiac disease, gluten-free diet, body mass index, clinical presentation

## Abstract

The association of clinical variables with body mass index (BMI) and changes experienced during a gluten-free diet (GFD) in celiac disease (CD) is not well established. In this retrospective cohort study, we aimed to investigate factors aligned with baseline and a follow-up regarding BMI in CD cases diagnosed at the University of Pécs (Hungary). Data were collected regarding gender, age, clinical presentation, histology, serology, extraintestinal manifestations, and BMI upon diagnosis and during follow-up. To compare variables with baseline BMI and BMI changes in short-, intermediate-, and long-term periods, we applied univariate analyses. A total of 192 CD patients were included. Males had significantly higher mean BMI when compared with females at diagnosis (22.9 ± 4.1 vs. 21.4 ± 4.3 kg/m^2^, *p* = 0.041) and during follow-up (*p* = 0.031, *p* = 0.029, and *p* = 0.033 for short-, intermediate-, and long-term follow-ups, respectively). Non-classical CD patients experienced higher mean BMI at diagnosis (22.9 ± 4.0 vs. 20.7 ± 4.4 kg/m^2^, *p* < 0.001) and following long-term follow-up (24.5 ± 3.2 vs. 22.6 ± 3.4 kg/m^2^, *p* = 0.039) than classical patients. In conclusion, although the mean BMI remained in the normal range, it increased significantly during follow-up, even at the short-term follow-up. This change was characteristic for non-classical cases and males on the long-term follow-ups.

## 1. Introduction

Celiac disease (CD) is a chronic, autoimmune systemic disorder triggered by gluten exposure in genetically vulnerable individuals [1]. CD is one of the most prevalent genetically determined disorders, affecting approximately 1% of the world population, with growing prevalence [2]. The role of gluten in the pathomechanism regarding CD is essential since classical villus atrophy and malabsorption develop with the consumption of gluten-containing cereals. Therefore, the only effective treatment for CD is a strict, lifelong gluten-free diet (GFD), excluding gluten proteins in wheat, barley, rye, and other related grains [3].

Current celiac guidelines usher in recommendations regarding dietary counseling upon diagnosis with the aim of maintaining a strict GFD and avoiding gluten contamination [4,5,6,7,8]. In most cases aligned with a well-managed diet, symptoms resolve, mucosa regenerates, celiac-specific antibodies normalize, and deficiency states are resolved [9]. Once absorption improves with diet, a CD patient’s body weight and body mass index (BMI) typically increase [10,11].

Recently, the clinical phenotype regarding CD has undergone a significant change: the number of non-classical or asymptomatic CD patients has increased [12]. Generally, the majority of untreated CD patients afflicted with classical clinical presentation are underweight and have lower BMI, fat mass (FM), fat-free mass (FFM), and bone mass, when compared to non-celiac counterparts [13,14]. However, patients with non-classical CD are not necessarily lean: they typically have normal body weight or can be even overweight or obese (with a BMI >25 kg/m^2^) [15]. Weight gain is desirable in the underweight yet not in those with normal or high body weights. Another problem surfaces when weight gain is mainly due to an increase in body FM rather than in FFM [16]. Thus, the result of a GFD can be unfavorable regarding body composition and nutrition-related disorders, such as non-alcoholic fatty liver disease (NAFLD) and cardiovascular (CV) events [17,18,19].

Determinants of nutritional status, especially BMI upon diagnosis of CD, and factors influencing BMI during a GFD are yet to be studied. In this study, we investigated the associations of BMI at diagnosis of CD and during the GFD, with a special focus on its clinical presentation.

## 2. Materials and Methods

This study is reported in conformity with the Strengthening the Reporting of Observational studies in Epidemiology (STROBE) Statement [20] (Appendix A). The study is conducted in full accordance with the Declaration of Helsinki and is approved by the Regional and Local Research Ethics Committee of University of Pécs, Pécs, Hungary (Ref. No. 6918).

### 2.1. Study Design

This single-centered, retrospective cohort study involves patients from our tertiary center enrolled and admitted at the University of Pécs (Pécs, Hungary). A portion of this study population was included in three previous studies reporting on different study goals and outcomes [21,22,23].

### 2.2. Study Population and Eligibility

The flow chart representing the selection of patients is presented in Figure 1. Patients were eligible for inclusion if they were diagnosed with CD in adulthood (≥18 years of age) and their BMI at diagnosis and during follow-up was available. All CD patients following diagnosis—verified by a gastroenterologist based on the combination of clinical, serological, and histopathological data as per the currently valid guidelines [4,5,6,7,8]—were instructed to follow a GFD.

### 2.3. Data Extraction

Data were extracted by using paper-based medical files and the current medical software, eMedSolution, based on disease identifiers. The time period for paper-based and electronic data collection begins with the years of 1992 and 2007, respectively, and concludes in 2022. Data of all eligible patients were manually retrieved by investigators with a healthcare degree into a pre-defined data collection table. Next, all data points were verified by a third investigator, also with a medical degree.

The following parameters were collected: gender, age upon diagnosis, calendar year of diagnosis of CD, clinical presentation (classical or non-classical CD by the Oslo Classification [24]), histology and serological results (anti-tissue transglutaminase (tTG) IgA and IgG, anti-endomysial antibody (EMA) IgA and IgG), the presence of IgA deficiency, anemia, osteoporosis, dermatitis herpetiformis, body height (m) and body weight (kg) upon diagnosis of CD, and body weight (kg) 1–15 years following diagnosis of CD.

Diagnostic histological samples (a minimum of four) were taken from the distal part of the duodenum. The samples were oriented, processed, and assessed by a gastrointestinal histopathologist.

A tTG level >10 U/mL was regarded positive (Orgentec Diagnostika GmbH, Mainz, Germany). Patients with high and low tTG titers were defined as ≥10 times or ˂10 times the upper limit of normal.

EMA results were declared positive when a reticular pattern of immunofluorescence was discerned in the muscular mucosae at a serum dilution ≥1:5.

IgA level was measured using nephelometry, and IgA deficiency was diagnosed based on the International Consensus definition as serum levels of IgA being below 7 mg/dL following the age of 4 years without IgG and IgM deficiency.

According to the WHO definition, anemia was established if hemoglobin levels were <12.0 g/dL in females and <13.0 g/dL in males.

To assess bone mineral density, dual-energy X-ray absorptiometry (DEXA) was applied (Horizon A (S/N 301472M), Hologic, Marlborough, MA, USA). Based on the WHO diagnostic criteria for osteoporosis, bone mineral density was considered normal when the T-score was −1.0 or greater, osteopenia occurs if T-score was between −2.5 and −1.0, and osteoporosis implies the T-score was −2.5 or less.

BMI was calculated for each year (weight divided by the square of height, given in kg/m^2^). BMI groups were created based on the World Health Organization (WHO) classification: (1) underweight—BMI under 18.5 kg/m^2^; (2) normal weight—BMI greater than or equal to 18.5 to 24.9 kg/m^2^; (3) overweight—BMI greater than or equal to 25.0 to 29.9 kg/m^2^; (4) obesity class I—BMI 30.0 to 34.9 kg/m^2^; (5) obesity class II—BMI 35.0 to 39.9 kg/m^2^; and (6) obesity class III—BMI greater than or equal to 40.0 kg/m^2^ [25].

### 2.4. Statistical Analysis

An expert biostatistician carried out the analyses using the software IBM SPSS Statistics 28 (IBM Corporation, Armonk, NY, USA).

Patients were allocated into three groups by length of follow-up: short- (1–2 years), intermediate- (3–5 years), and long-term follow-up (6–15 years), corresponding to the length of the GFD. Variables of interest were analyzed across these groups.

Continuous variables were expressed as mean ± standard deviation if they showed a normal distribution, verified by the Kolmogorov–Smirnov test. In support of comparative analysis, with normal distribution, the independent *t*-test, the ANOVA test with Tukey’s HSD post hoc test, and the Repeated Measures ANOVA test were used. Regarding comparative analysis, with non-normal distribution, the non-parametric tests as the Mann–Whitney U-test and the Friedman test were used.

Categorical variables were expressed as relative frequencies. The data were analyzed using contingency tables and the Chi-squared or Fischer’s test, as appropriate.

Statistical significance was established as a *p*-value of < 0.05.

Post hoc power analysis was performed using G*Power 3.1 statistical power analysis software (Heinrich Heine University Düsseldorf, Düsseldorf, Germany) [26,27].

## 3. Results

A total of 192 CD patients were included in the study. Characteristics of the patients included are summarized in Table 1. Mean age upon diagnosis of CD was 37.5 ± 13.2 years (range: 18.0–70.0 years), and approximately three-fourths of patients were females. Out of the 192 subjects, 101 (52.6%) had classical CD. The mean BMI of the study population was 21.7 ± 4.3 kg/m^2^. Roughly half (50.5%) of the patients had normal body weight at diagnosis, followed by underweight and overweight classes. Most of the patients had normal IgA levels, no anemia, and osteoporosis nor dermatitis herpetiformis upon diagnosis.

### 3.1. Clinical Variables and Mean BMI at Diagnosis of CD

In terms of gender, males had significantly higher mean BMI upon diagnosis of CD than when compared with females (22.9 ± 4.1 vs. 21.4 ± 4.3 kg/m^2^, respectively, *p* = 0.041).

Concerning anthropometric parameters, non-classical CD patients had significantly higher mean BMI upon diagnosis than when compared with classical CD patients (*p* < 0.001). The majority of classical and non-classical CD patients belonged to the normal BMI class. The proportion of underweight patients was significantly lower, and the proportion of overweight patients was significantly higher in non-classical cases than classical CD (*p* < 0.001 for both) (Table 1).

Mean BMI upon diagnosis of CD was significantly associated with the tTG IgG titer category (*p* = 0.024), meaning those patients with high positive tTG IgG titers had significantly lower mean BMI, when compared to patients with negative titers (20.2 ± 3.4 vs. 22.9 ± 4.7 kg/m^2^, respectively, *p* = 0.026).

Mean BMI upon diagnosis of CD did not significantly differ by other variables including serological results (tTG IgA, EMA titers), histology, IgA deficiency, anemia, osteoporosis, and dermatitis herpetiformis at diagnosis (*p* > 0.05 for all comparisons).

### 3.2. Clinical Variables and BMI Classes at Diagnosis of CD

Factors significantly associated with the BMI class at a diagnosis of CD are presented in Table 2.

With respect to gender, a significant difference was observed in the underweight and overweight classes: males were more likely to be overweight than when compared with females (34.9% vs. 12.1%), whereas females were more likely to be underweight than males (30.2% vs. 16.3%) (*p* = 0.010 for interaction).

There was a statistically significant association between clinical presentation and BMI class (*p* < 0.001 for interaction). As expected, the proportion of underweight patients was higher with classical CD compared to non-classical CD (39.6% vs. 13.2%, respectively), whereas overweight patients were more likely to have non-classical CD (24.2% vs. 10.9%, respectively).

Those who had anemia tended to be underweight (36.6% of the cases), whereas among those without anemia, only 20.0% of patients were underweight. In contrast, the proportion of overweight patients was higher in patients without anemia, compared to those with anemia (21.0% vs. 12.2%, respectively). There was a significant association between the BMI class and presence of anemia (*p* = 0.035 for interaction).

Data on IgA deficiency, serology, histology, osteoporosis, and dermatitis herpetiformis at diagnosis of CD did not significantly differ across BMI classes (*p* > 0.05 for all comparisons).

### 3.3. Mean BMI Change during Follow-Up

As illustrated in Figure 2, the mean BMI at short-, intermediate-, and long-term follow-ups was significantly higher than when calculated at diagnosis (*p* < 0.001 for all comparisons).

Out of the 192 patients, only 17 had available BMI data for each time frame of follow-up. For characteristics of these patients, see Appendix A. Mean BMI increased significantly at short-term follow-up (*p* = 0.034), and BMI continued to rise during both intermediate- and long-term follow-ups, but only moderately (*p* > 0.05 for both). When comparing mean BMI at diagnosis to those calculated during follow-up, a significantly higher BMI was detected in all comparisons (Figure 3).

### 3.4. BMI Class Change during Follow-Up

Out of the 192 patients, only 17 had BMI classes for each time frame. The Friedman test showed a significant difference between the groups (*p* < 0.001). In pairwise comparisons, the proportion of higher BMI classes increased during follow-up (*p* = 0.034 for the comparisons of BMI at diagnosis of CD vs. that at intermediate- and long-term follow-ups) (Appendix A).

### 3.5. Association of BMI during Follow-Up with Clinical Variables

Males had a significantly higher mean BMI than females at all time frames (*p* = 0.031, *p* = 0.029, and *p* = 0.033 for short-, intermediate-, and long-term follow-ups, respectively).

In comparing the mean BMI of different time intervals, they differ significantly from BMI upon diagnosis in both males and females (Table 3).

Males upon diagnosis vs. those at short-term (Table 3) and short- vs. those at intermediate-term (n = 16, 25.5 ± 4.5 vs. 26.7 ± 4.8 kg/m^2^, respectively, *p* = 0.002) had a significantly lower mean BMI; however, intermediate- vs. long-term comparisons did not yield a statistically significant difference. In females, a statistically significant difference in mean BMI was observed only upon diagnosis vs. those at short-term follow-up (Table 3); however short- vs. intermediate-term and intermediate- vs. long-term comparisons did not yield a statistically significant difference.

Considering the association between clinical presentation and BMI, the significant difference observed at diagnosis was not detectable at short- and intermediate-term follow-ups; however, at the long-term follow-up, patients with non-classical CD (n = 25) had a higher BMI, compared to those with classical CD (n = 36) (24.5 ± 3.2 vs. 22.6 ± 3.4 kg/m^2^, respectively, *p* = 0.039).

When comparing BMI change (from the diagnosis of CD) by clinical presentation, there was a significant difference between classical and non-classical CD patients at all time frames (Table 4).

Regarding serology upon diagnosis of CD, significant short-term BMI change was detected between the tTG IgA titer groups (*p* = 0.005) and tTG IgG titer (*p* = 0.038) groups for CD patients. Concerning tTG IgA, a significantly higher BMI change was observed between cases with low positive and high positive titer (*p* = 0.008) when compared to those with negative vs. low positive and negative vs. high positive titer. Concerning tTG IgG, cases with negative vs. low positive titer had a significantly higher BMI change (*p* = 0.033), compared to those with negative vs. high positive and low positive vs. high positive titer. Upon intermediate- and long-term follow-ups, there was no significant association between BMI change and serology.

Data on other examined variables were not significantly associated with BMI change.

### 3.6. Post Hoc Power Analysis

Power analysis was performed at all groups where short-, intermediate-, and long-term BMIs were compared to BMI upon diagnosis, as in each group, different patient numbers were available. In all cases, the power of the test was above the 0.8 value usually requested by researchers (Table 5).

For all time frames of follow-up, 17 patients had available BMI data. In this aspect, the power of the test calculated with the G*Power software was 0.73, which is slightly below the generally required value of 0.8.

## 4. Discussion

Classically, CD is a potential cause of malnutrition since it is characterized by intestinal villous atrophy, which leads to malabsorption. In typical cases, CD patients are malnourished and exhibit deficiency symptoms. However, a proportion of patients—despite the presence of villous atrophy—have no clinical signs and laboratory abnormalities relating to malabsorption or have only isolated abnormalities (e.g., anemia). These patients are typically recognized by extraintestinal manifestations of the disease (e.g., dermatitis herpetiformis, osteoporosis, or liver function test abnormalities). With the improvement of diagnostics and disease awareness, the fraction of non-classical CD cases is increasing, and today, it has become more prevalent than the classical forms [28,29,30,31,32]. According to our previous study, this trend is also observed among our CD patients [23]; whereas, in the present study, the classical presentation of CD was slightly more frequent (52.6%).

A recent review found all anthropometric parameters to be worse in untreated CD patients when compared to controls [33]. Depending on geographical regions, considerable differences exist in the proportion of under- and overweight CD patients. In an Indian study, the proportion of underweight CD patients was 36.2% [15], contrasting to an Italian study and a Finnish study, in which it was only 6% and 4%, respectively [34,35]. In consideration of a study originating in the US, the mean BMI among classical CD patients was relatively high (24.4 kg/m^2^), yet those of non-classical cases was even higher (25.7 kg/m^2^) [36]. This tendency is supported by another study from the US, in which nearly half of the CD patients were already obese (BMI >30 kg/m^2^) upon diagnosis, and the prevalence of obesity continued to rise linearly over the 5-year study span between 2014 and 2018 [37]. The prevalence of obesity is increasing worldwide, and this tendency is also true among CD patients. Tucker et al. demonstrated that 44% of CD patients were overweight and 13% were obese upon diagnosis, and, over the years of the GFD, the proportion of both classes increased steadily [38]. A Chilean retrospective study reported similar results: the later the calendar year of diagnosis, the better the nutritional status and the higher the proportion of obese CD patients [17].

In our study, the proportion of underweight patients upon diagnosis was relatively high (27.1%), which can be partially explained by the predominance of classical CD cases. A significant difference was observed by gender: although the most common BMI class was the normal in both genders, in which a vast proportion of the females were underweight, males tended to be overweight. This difference can only partially be explained by the more common presence of the classical presentation among females (53.7% vs. 48.8%) than males. What is more important is that males had a prominent rise in BMI at the intermediate time frame of follow-up, moving them from the normal to the overweight class. This change is unfavorable from several points of view: it further increases the already high CV risk in aging males and—especially when combined with alcohol consumption—exacerbates the risk of developing fatty liver disease.

The association of clinical presentation and other diagnostic features of CD with BMI has not been previously studied. In our study, CD patients with classical presentation and high positive tTG IgG titers had a significantly lower BMI upon diagnosis than non-classical cases. Data on BMI at diagnosis did not significantly differ by IgA deficiency, other serological results (tTG IgA, EMA titers), histology, anemia osteoporosis, and dermatitis herpetiformis upon diagnosis. In consideration of BMI classes upon diagnosis, patients with anemia tended to be underweight (36.6%), whereas a significant fraction of patients without anemia were overweight (21.1%). The context of a low BMI upon diagnosis with classical symptoms and high antibody titers is not surprising: a more prominent immune response assumes more severe mucosal damage and malabsorption.

Based on recent reviews, following a GFD resulted in generally improved nutritional status and significantly increased BMI [19,33,36,39]. Our study supports this, showing the extent of BMI changes was significantly higher among patients with classical CD following a GFD in short-, intermediate-, and long-term follow-ups and in the short-term for both those with high tTG IgA and tTG IgG titers. The reason may be that patients with severe malabsorption and high antibody counts showed the most marked improvement in absorption due to the diet, since they had to catch up to normal nutritional status.

Weight gain due to the GFD is desirable for some patients but not always beneficial for others. Several studies draw attention to GFD-induced weight gain, which was associated with disproportionate body composition, as it resulted in a substantial gain in FM and a modest increase in FFM [13,16,40]. In an Irish study, during a 2-year follow-up, weight gain occurred in 81%, whereas those who were initially overweight continued to gain weight and the proportion of overweight individuals increased from 26% to 51% [41]. Another retrospective study from Israel among adult CD patients reported the BMI among individuals following a GFD significantly increased, hence the proportion of overweight/obese (BMI ≥25 kg/m^2^) patients rose from 32% to 38.8% [42]. It is probable the composition of gluten-free products is generally less favorable than their gluten-containing counterparts since it has high calorie, saturated fat, simple carbohydrate, and sugar syrup contents [43,44].

The metabolic consequences of increased body weight and FM are most likely to occur in patients who are not underweight upon diagnosis of CD. The increased incidence of NAFLD in CD has been demonstrated by several studies [9,18,45]. Surprisingly, the prevalence of NAFLD is already higher in untreated CD patients compared to the general population (hazard ratio (HR): 2.8, 95% CI 2.0–3.8). The risk increasing in the first year following CD diagnosis was 13.3 (95% CI 3.5–50.3) but remained significantly elevated even beyond 15 years following the diagnosis of CD (HR = 2.5; 95% CI 1.0–5.9) [45]. Tortora et al. presented the proportion of metabolic syndrome among newly diagnosed CD patients was 2%, rising to 30% following 1 year of a GFD [9].

Strengths of our study comprise the consecutive patient involvement. The association between BMI and clinical presentation has not yet been investigated. Limitations of our study include its retrospective nature, variation in data recording with time, and limited available data on follow-up BMI. The density of the dataset did not allow us to perform multiple regression analysis. Additionally, the adherence to the GFD was not evaluated and systematically documented.

## 5. Conclusions

In our study, the proportion of underweight patients upon diagnosis was relatively high, especially in females. For these patients, the main challenge remains the elimination of malnutrition and the improvement of the nutritional status. According to our results, the BMI upon diagnosis was significantly higher in males and in patients with a non-classical phenotype. Although the mean BMI was within the normal range before and after a GFD, the mean BMI already increased significantly on the short-term follow-up. The significant difference in mean BMI observed upon diagnosis between classical and non-classical cases disappeared on the short-term with a “catch up weight gain” in the classical group, but on the long-term, a significant difference was observed again with a BMI increase in the non-classical group. The number of the underweight patients tended to decrease with time during a GFD, and even on the short-term, the number of the overweight, mainly affecting males and non-classical cases, increased on the long-term. For the management of obesity-related problems, CV and metabolic consequences are new challenges in CD patient care. Nevertheless, dietary counseling and regular monitoring play important roles in ensuring adverse effects of a GFD do not prevail and weight gain is proportionate, as primarily FFM and not FM increases, and these interventions should particularly focus on non-classical cases and males.

## Figures and Tables

**Figure 1 nutrients-15-03517-f001:**
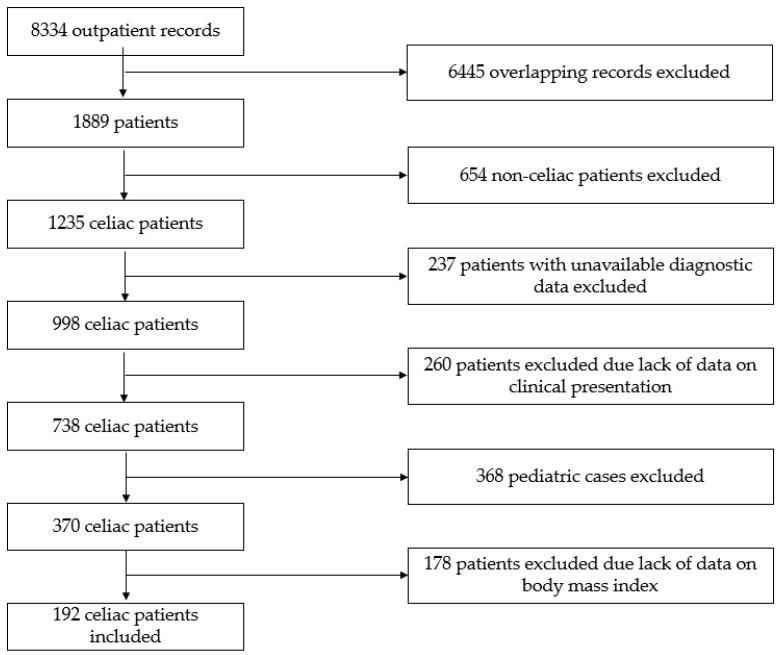
Flow chart of the study.

**Figure 2 nutrients-15-03517-f002:**
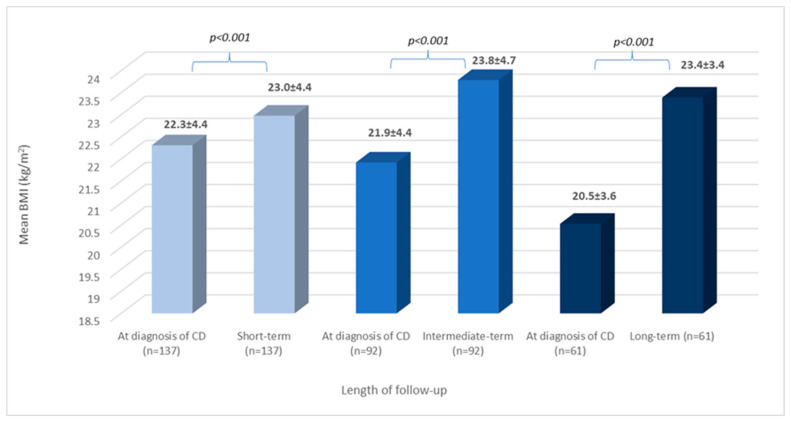
BMI change during follow-up: all patients. BMI: body mass index; CD: celiac disease; n: number of patients; and values are reported in mean and standard deviation: x ± SD.

**Figure 3 nutrients-15-03517-f003:**
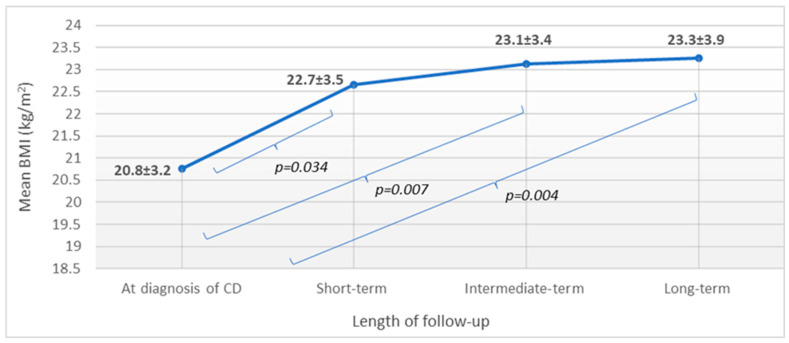
BMI change during follow-up: patients having available data for all time frames (n = 17). n: number of patients; BMI: body mass index; CD: celiac disease; and values are reported in mean and standard deviation: x ± SD.

**Table 1 nutrients-15-03517-t001:** Characteristics of patients included.

Variables	Total Cohort of Patients	Patients with Classical CD	Patients with Non-Classical CD	*p*-Value (Classical vs. Non-Classical CD)
Age at diagnosis	N = 192	N = 101	N = 91	
Mean ± standard deviation, years	37.5 ± 13.2	37.9 ± 14.1	37.0 ± 12.1	0.638
Gender	N = 192	N = 101	N = 91	
Males	43 (22.4%)	21 (48.8%)	22 (51.2%)	0.574
Females	149 (77.6%)	80 (53.7%)	69 (46.3%)	
BMI at diagnosis	N = 192	N = 101	N = 91	
Mean ± standard deviation, kg/m^2^	21.7 ± 4.3	20.7 ± 4.4	22.9 ± 4.0	<0.001 *
Underweight	52 (27.1%)	40 (39.6%)	12 (13.2%)	<0.001 *
Normal	97 (50.5%)	45 (44.6%)	52 (57.1%)	
Overweight	33 (17.2%)	11 (10.9%)	22 (24.2%)	
Obesity class I	9 (4.7%)	4 (4.0%)	5 (5.5%)	
Obesity class II	1 (0.5%)	1 (1.0%)	0 (0.0%)	
Obesity class III	0 (0.0%)	0 (0.0%)	0 (0.0%)	
Histology at diagnosis	N = 129	N = 70	N = 59	
Marsh 1–2	6 (4.7%)	3 (50.0%)	3 (50.0%)	0.718
Marsh 3a-3b	54 (41.9%)	27 (50.0%)	27 (50.0%)	
Marsh 3c	69 (53.5%)	40 (58.0%)	29 (42.0%)	
tTG IgA at diagnosis	N = 172	N = 89	N = 83	
Negative	12 (7.0%)	6 (50.0%)	6 (50.0%)	0.948
Low positive	58 (33.7%)	31 (53.4%)	27 (46.6%)	
High positive	102 (59.3%)	52 (51.0%)	50 (49.0%)	
tTG IgG at diagnosis	N = 158	N = 80	N = 78	
Negative	66 (41.8%)	32 (48.5%)	34 (51.5%)	0.878
Low positive	68 (43.0%)	35 (51.5%)	33 (48.5%)	
High positive	24 (15.2%)	13 (54.2%)	11 (45.8%)	
EMA IgA at diagnosis	N = 142	N = 69	N = 73	
Negative	19 (13.0%)	9 (47.4%)	10 (52.6%)	0.619
Weak positive	13 (9.2%)	8 (61.5%)	5 (38.5%)	
Strong positive	110 (77.5%)	52 (47.3%)	58 (52.7%)	
EMA IgG at diagnosis	N = 95	N = 47	N = 48	
Negative	35 (36.8%)	20 (57.1%)	15 (42.9%)	0.520
Weak positive	8 (8.4%)	4 (50.0%)	4 (50.0%)	
Strong positive	52 (54.7%)	23 (44.2%)	29 (55.8%)	
IgA deficiency at diagnosis	N = 92	N = 44	N = 48	
No	86 (93.5%)	41 (47.7%)	45 (52.3%)	1.000
Yes	6 (6.5%)	3 (50.0%)	3 (50.0%)	
Anemia at diagnosis	N = 187	N = 99	N = 88	
No	105 (56.1%)	51 (48.6%)	54 (51.4%)	0.176
Yes	82 (43.9%)	48 (58.5%)	34 (41.5%)	
Bone mineral density at diagnosis	N = 110	N = 63	N = 47	
No	47 (42.7%)	25 (53.2%)	22 (46.8%)	0.184
Osteopenia	32 (29.1%)	16 (50.0%)	16 (50.0%)	
Osteoporosis	31 (28.2%)	22 (71.0%)	9 (29.0%)	
Dermatitis herpetiformis at diagnosis	N = 192	N = 101	N = 91	
No	166 (86.5%)	92 (55.4%)	74 (44.6%)	0.048 *
Yes	26 (13.5%)	9 (34.6%)	14 (65.4%)	

N: number of patients; tTG: anti-tissue transglutaminase antibody; EMA: anti-endomysial antibody; BMI: body mass index; and * indicates statistical significance. CD: celiac disease; Dichotomous variables are expressed in number of patients and percentage of total.

**Table 2 nutrients-15-03517-t002:** Factors significantly associated with BMI class at diagnosis of CD.

Variables		Total Number of Patients	BMI Class at Diagnosis						*p*-Value
			Underweight	Normal Weight	Overweight	Obesity Class I	Obesity Class II	Obesity Class III	
Gender	Males	43	7 (16.3%)	19 (44.2%)	15 (34.9%)	2 (4.7%)	0 (0.0%)	0 (0.0%)	0.010 *
	Females	149	45 (30.2%)	78 (52.3%)	18 (12.1%)	7 (4.7%)	1 (0.7%)	0 (0.0%)	
Clinical presentation	Classical	101	40 (39.6%)	45 (44.6%)	11 (10.9%)	4 (4.0%)	1 (1.0%)	0 (0.0%)	<0.001 *
	Non-classical	91	12 (13.2%)	52 (57.1%)	22 (24.2%)	5 (5.5%)	0 (0.0%)	0 (0.0%)	
Anemia	No	105	21 (20.0%)	58 (55.2%)	22 (21.0%)	3 (2.9%)	1 (1.0%)	0 (0.0%)	0.035 *
	Yes	82	30 (36.6%)	37 (45.1%)	10 (12.2%)	5 (6.1%)	0 (0.0%)	0 (0.0%)	

BMI: body mass index; CD: celiac disease; EMA: anti-endomysial antibody; * presented: significant result; and values are reported in relative frequency and percentage.

**Table 3 nutrients-15-03517-t003:** BMI within genders during follow-up.

Length of Follow-Up	Gender	N	Mean BMI (kg/m^2^)	*p*-Value
At diagnosis of CD	males	29	23.1 ± 4.1	0.042 *
Short-term follow-up			24.5 ± 4.3	
At diagnosis of CD		25	23.4 ± 4.0	0.001 *
Intermediate-term follow-up			25.5 ± 4.6	
At diagnosis of CD		9	22.4 ± 4.1	0.015 *
Long-term follow-up			25.6 ± 2.3	
At diagnosis of CD	females	108	21.9 ± 4.4	0.001 *
Short-term follow-up			22.6 ± 4.3	
At diagnosis of CD		67	21.4 ± 4.4	<0.001 *
Intermediate-term follow-up			23.1 ± 4.6	
At diagnosis of CD		52	20.2 ± 3.4	<0.001 *
Long-term follow-up			23.0 ± 3.4	

N: number of patients; BMI: body mass index; CD: celiac disease; values are reported in mean and standard deviation: x ± SD; and * presented: significant result.

**Table 4 nutrients-15-03517-t004:** BMI change by clinical presentation of CD.

Length of Follow-Up	Clinical Presentation	N	Mean BMI Change (kg/m^2^)	*p*-Value
Short-term follow-up	classical	64	+ 1.1 ± 2.3	0.029 *
	non-classical	73	+ 0.3 ± 1.7	
Intermediate-term follow-up	classical	48	+ 2.9 ± 3.4	<0.001 *
	non-classical	44	+ 0.7 ± 2.4	
Long-term follow-up	classical	36	+ 3.6 ± 2.8	0.007 *
	non-classical	25	+ 1.8 ± 2.0	

N: number of patients; BMI: body mass index; + indicates the increase in BMI; CD: celiac disease; values are reported in mean and standard deviation: x ± SD; and * presented: significant result.

**Table 5 nutrients-15-03517-t005:** Results of the post hoc power analysis.

Length of Follow-Up	N with Available BMI	Power of the Test
At diagnosis of CD	137	0.98
Short-term follow-up		
At diagnosis of CD	92	0.99
Intermediate-term follow-up		
At diagnosis of CD	61	0.99
Long-term follow-up		

N: number of patients; BMI: body mass index; and CD: celiac disease.

## Data Availability

The data presented in this study are available within the article. The raw data are available on request from the corresponding author.

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
