# Peer review of "Body Mass Index during Gluten-Free Diet in Patients with Celiac Disease"

_nutrients, 2023, doi:10.3390/nu15163517_

Round 1

Reviewer 1 Report

This is an interesting study aimed to investigate factors associated with BMI in CD patients. Mean BMI at short-, intermediate-, and long-term follow-up was significantly higher than that calculated at diagnosis. The ratio of underweight CD patients tended to decrease with time during GFD. The authors gave detailed description of the obtained results. However, some major revisions are needed.

- In the Material and Methods section, the “Power analysis and sample size calculation” paragraph should be added, before statistical analysis paragraph.

- A multiple regression could be performed after correlation test.

- Line 161: The authors state that “men were more likely to be overweight than women” but the BMI are not reported. It would be an interesting information to better understand the magnitude of the difference. Please add this data also for classical vs non-classical CD.

- Line 197-198: “Out of the 192 patients, only 17 had BMI classes for each time period. The Friedman test showed a significant difference between the groups (p<0.001).” An assumption of Friedman test is that the variable is measured on three or more different occasions. How many patients satisfied this condition?

- Line 314: The authors state “Strengths of our study comprises its sample size”. However, out of the 192 patients, only 17 had available BMI data for each time frame of follow-up. Since the aim of the study was to investigate factors being associated with baseline and follow-up BMI, which was the sample size calculated to reach the statistical power?

- The Ethical clearance paragraph is lacking. Please add the Ethical Committee approval ID.

Author Response

This is an interesting study aimed to investigate factors associated with BMI in CD patients. Mean BMI at short-, intermediate-, and long-term follow-up was significantly higher than that calculated at diagnosis. The ratio of underweight CD patients tended to decrease with time during GFD. The authors gave detailed description of the obtained results. However, some major revisions are needed.

Thank you for your efforts spent on reviewing our paper. We appreciate your valuable comments.

Comment 1: In the Material and Methods section, the “Power analysis and sample size calculation” paragraph should be added, before statistical analysis paragraph.

Response 1: Thank you for the suggestion. In our retrospective study, we enrolled all CD patients who fulfilled the inclusion criteria during the study period (1992-2022). In this setting, we were unable to perform sample size calculation a priori as the number of CD patients was predetermined by the number of cases attending our clinic. Due to the retrospective setting, we decided not to perform power calculation because irrespective of the results, we are unable to recruit further patients.

Comment 2: A multiple regression could be performed after correlation test.

Response 2: Thank you for your comment. The density of the dataset did not allow us to perform multiple regression analysis (see, limitations in the Discussion section).

Comment 3: Line 161: The authors state that “men were more likely to be overweight than women” but the BMI are not reported. It would be an interesting information to better understand the magnitude of the difference. Please add this data also for classical vs non-classical CD.

Response 3: Fair point, however this paragraph discusses BMI classes and not mean BMI values. We added relative frequencies to the corresponding section (Pg. 6, Ln. 175-176).

Comment 4: Line 197-198: “Out of the 192 patients, only 17 had BMI classes for each time period. The Friedman test showed a significant difference between the groups (p<0.001).” An assumption of Friedman test is that the variable is measured on three or more different occasions. How many patients satisfied this condition?

Response 4: All the 17 patients fulfilled this condition. In case of Friedman test, we calculated the effect size statistic, Kendall’s w was 0.68, so it means that more than 0.5, which is the borderline value of the large effect.

Comment 5: Line 314: The authors state “Strengths of our study comprises its sample size”. However, out of the 192 patients, only 17 had available BMI data for each time frame of follow-up. Since the aim of the study was to investigate factors being associated with baseline and follow-up BMI, which was the sample size calculated to reach the statistical power?

Response 5: Thank you for the comment. We removed the corresponding sentence from the manuscript because neither a priori sample size calculation nor post-hoc power calculation was performed (as detailed under Comment 1). An important limitation of this work is that in some analysis, data availability was limited (see the section about strengths and limitations in the manuscript).

Comment 6: The Ethical clearance paragraph is lacking. Please add the Ethical Committee approval ID.

Response 6: Please, see Pg. 2, Ln. 68-73 and the Institutional Review Board Statement, Pg. 11, Ln. 364-366.

Reviewer 2 Report

The manuscript of Vereczkei Z. et al. "Celiac Men and Patients with Non-Classical Celiac Disease Are Vulnerable to Be Overweight During Gluten-Free Diet" is interesting and concerns very important topic. Database used is rich as for quite limited evidence of celiac disease in general.

I see 2 major points that need to be reconsidered in this manuscript.

1. The title and general concept. As mean BMI in patients even after follow-up is within the normal range I think it is not correct to say that patients tend to be overweight. I would rather consider that they have probably milder symptoms and tend to lose less weight if compared to those with classical symptoms. To make conclusion of tendency to be overweight different approach should be selected. This requires comparison with healthy peers. This needs to be revised throughout the manuscript.

2. I strongly recommend  a native speaker of English to review the language.

Text must be reviewed by (preferably) native speaker 

Author Response

The manuscript of Vereczkei Z. et al. "Celiac Men and Patients with Non-Classical Celiac Disease Are Vulnerable to Be Overweight During Gluten-Free Diet" is interesting and concerns very important topic. Database used is rich as for quite limited evidence of celiac disease in general. I see 2 major points that need to be reconsidered in this manuscript.

Thank you for your efforts spent on reviewing our paper. We appreciate your valuable comments.

Comment 1: The title and general concept. As mean BMI in patients even after follow-up is within the normal range, I think it is not correct to say that patients tend to be overweight. I would rather consider that they have probably milder symptoms and tend to lose less weight if compared to those with classical symptoms. To make conclusion of tendency to be overweight different approach should be selected. This requires comparison with healthy peers. This needs to be revised throughout the manuscript.

Response 1: We agree with your statement. Accordingly, we attempted to rephrase the title of the manuscript and revised the whole manuscript from this point of view. We reformulated the conclusion as well.

Comment 2: I strongly recommend a native speaker of English to review the language.

Response 2: A native speaker of English revised and improved the language of the paper.

Round 2

Reviewer 1 Report

I would like to thank the authors for the answers provided. Comments 2, 3, 5, and 6 were fully addressed. Responses 1 and 4 should be further discussed.

Response 1: Thank you for the suggestion. In our retrospective study, we enrolled all CD patients who fulfilled the inclusion criteria during the study period (1992-2022). In this setting, we were unable to perform sample size calculation a priori as the number of CD patients was predetermined by the number of cases attending our clinic. Due to the retrospective setting, we decided not to perform power calculation because irrespective of the results, we are unable to recruit further patients.

It is not the issue of the design, retrospective or prospective, but the study hypotheses. You should perform power calculation on the basis of your research question, then if you get the mimimum number in your available data you are fine.

Response 4: All the 17 patients fulfilled this condition. In case of Friedman test, we calculated the effect size statistic, Kendall’s w was 0.68, so it means that more than 0.5, which is the borderline value of the large effect.

As reported by the authors, the main aim was “to investigate factors aligned with baseline and a follow-up regarding BMI in CD cases”, so telling about 192 patients while only 17 patients had available BMI data for each time frame of follow-up could be misleading. The paper should be on 17 patients, maybe as Short Communication.

Reviewer 2 Report

Some minor errors still remain:

Line 2 Title still not perfect. ...diet in PATIENTS with celiac disease...

Line 15: ... in celiac disease (CD) is still unclear -> is not well established or is not well studied or similar. 

Line 331 femles-> females.

text should be checked for spelling errors and minor problems

Round 3

Reviewer 2 Report

no additional comments

acceptable